# Preparation of Monoclonal Antibody against Pyrene and Benzo [a]pyrene and Development of Enzyme-Linked Immunosorbent Assay for Fish, Shrimp and Crab Samples

**DOI:** 10.3390/foods11203220

**Published:** 2022-10-15

**Authors:** Shuangmin Wu, Huaming Li, Xiaoyang Yin, Yu Si, Liangni Qin, Hongfei Yang, Jiaxu Xiao, Dapeng Peng

**Affiliations:** National Reference Laboratory of Veterinary Drug Residues (HZAU) and MOA Key Laboratory for the Detection of Veterinary Drug Residues in Foods, Huazhong Agricultural University, Wuhan 430070, China

**Keywords:** pyrene, benzo [a]pyrene, monoclonal antibody, enzyme-linked immunoassay, aquatic product

## Abstract

Polycyclic aromatic hydrocarbons (PAHs) are significant environmental and food pollutants that can cause cancer. In this work, a specific monoclonal antibody (mAb) to identify pyrene (PYR) and benzo [a]pyrene (BaP) was prepared, and an indirect competitive enzyme-linked immunoassay (ic-ELISA) was established to detect PYR and BaP residues in living aquatic products for the first time. The effects of complete antigens with different coupling ratios on the production of high-sensitivity mAb was explored. Under the optimal conditions, the IC_50_ value was 3.73 ± 0.43 µg/L (n = 5). The limits of detection (LODs) for PYR and BaP in fish, shrimp, and crab ranged from 0.43 to 0.98 µg/L. The average recoveries of the spiked samples ranged from 81.5–101.9%, and the coefficient of variation (CV) was less than 11.7%. The validation of the HPLC-FLD method indicated that the ELISA method set up in this experiment provided a trustworthy tool for PAHs residues detection in aquatic products.

## 1. Introduction

PAHs, composed of two or more aromatic rings, are the most ubiquitous persistent organic hydrocarbons in the world [1]. The chemistry properties of PAHs (hydrophobicity, stability and carcinogenicity) will change as the ring number of PAHs increases [2,3]. The main source of PAHs is the incomplete combustion of organic materials such as fossil fuels, natural gas and wood, as well as human activities, including food processing, garbage incineration and automobile exhaust fumes [4,5,6]. PAHs can migrate with air and water for a long distance and accumulate in organisms, causing extensive and serious harm to human fitness and the ecological environment. Exposure to PAHs can cause a variety of adverse effects in humans and animals, including carcinogenicity, DNA damage, teratogenicity, mutagenicity and immunotoxicity [7,8,9,10]. Therefore, 16 PAHs have been classified as significant pollutants by the European Scientific Committee for Food (ECSCF) [11].

The pollution of PAHs in water environments is mainly caused by wastewater discharge and oil spills. For example, in 2010, the famous oil spill occurred in the Gulf of Mexico, and millions of tons of oil were released into the Gulf of Mexico [12]. China is a large aquaculture country, and its aquaculture production makes up more than 69% of the total global production. However, monitoring of major pollution sources in the national water environment in 2018 showed that PAHs were found in water, sediment and aquatic organisms. With the significant increase in industrial production and demand, the level of PAH-based pollutants in aquatic ecosystems has become strikingly high [13]. Owing to PAHs lipophilicity, they can easily accumulate in aquatic organisms and cause serious harm because their membranes are easily penetrated by PAHs [14]. It was reported that PAHs tend to accumulate in fatty aquatic products such as fish, shrimp and crabs. PAHs can interfere with estrogen signaling pathways (especially 4- and 5-ring) [15,16]. PYR and BaP (Appendix A) are the main tetracyclic and pentacyclic compounds in PAHs, with strong persistence and harmfulness in the water environment. The residues in aquatic products are important factors affecting the quality and safety of aquaculture [17]. BaP is a residue marker of PAH pollution exposure due to its high carcinogenicity [18]. The European Commission has set limits of 2.0 µg/kg for fish, 5.0 µg/kg for crustaceans and cephalopods, and 10.0 µg/kg for shellfish [19].

Many chromatographic methods have been developed to monitor PAHs in different matrices, including GC–MS, LC-MS, and HPLC [20,21,22]. Although chromatographic methods have some advantages, the sample pretreatment is complicated and requires professional operators, making it unsuitable for on-site testing of a large number of samples. Immunoassays are more desirable because of their high sensitivity, time savings and low requirements for operators [23]. At present, ELISA is a widely used detection method in immunoassay, which is simple, fast and greatly improves the efficiency of residue analysis [24]. However, there are few studies on the ELISA methods of both PYR and BaP, and they mainly focus on the detection of environmental samples, such as water (lake water, tap water, drinking water) and air [25]. As far as we know, there is no reported immunoassay method for the PAHs residues assay in aquatic products.

In this work, a highly sensitivity mAb was prepared and an ELISA method was developed to detect PYR and BaP residues in fish, shrimp and crab without complex sample pretreatment. The effect of organic solvents on the sensitivity of ELISA was investigated. The accuracy of the ELISA is reliable compared to HPLC-FLD.

## 2. Materials and Methods

### 2.1. Chemicals and Reagents

Seventeen standard analytes of PAHs, namely, naphthalene, pyrene, fluorene, chrysene, benzo(k)fluoranthene, benzo(g, h, i) perylene, acenaphthylene, dibenzo(a, h)anthracene, acenaphthene, benzo(a)pyrene, phenanthrene, indeno(1, 2, 3-cd)pyrene, benzo(a)anthracene, fluoranthene, benzo(b)fluoranthene, anthracene and pyrene butyric acid, were purchased from Dr. Ehrenstorfer GmbH (Augsbury, Germany). Peroxidase-labelled goat anti-mouse immunoglobulins (HRP-IgG), OVA, serum-free cell freezing medium, Freund’s complete adjuvant (FCA), BSA, hypoxanthine-aminopterin-thymidine (HAT), Freund’s incomplete adjuvant (FIA) and PEG1450 were bought from Sigma (St. Louis, MO, USA). All other chemicals and organic solvents used were obtained from reagent grade or better. 

### 2.2. Synthesis of Antigens

The synthesis method of the coating antigen and immunogen was appropriately improved by Meng et al. (2015) [15]. First, pyrene butyric acid (PBA) (110 mg), EDC (130 mg) and NHS (210 mg) were dissolved in 3 mL DMF. The solution was gently mixed at 4 °C overnight, which was called activated solution. BSA (70 mg) was dissolved in 9 mL of PBS (0.01 mol/L, pH = 7.4). Preparation of immunogen (PBA-BSA) with different coupling ratios: three glass bottles were used, and 3 mL BSA solution was added, respectively. Then, 400 µL, 600 µL, and 800 µL of activation solution was slowly added to the bottle successively and stirred at 4 °C for 10 h away from light. Three immunogens were represented by A_1_, A_2_ and A_3_.

The coating antigen (PBA-OVA) was prepared using the same method as the immunogen. Briefly, OVA (90 mg) was dissolved in 12 mL PBS, the above activated solution (200 µL, 300 µL, 400 µL) was added to three glass bottles containing 4 mL OVA protein solution. These solutions were gently stirred at 4 °C for 12 h. Three coating antigens were represented by B1, B2 and B3. They were exhaustively dialyzed with 0.1M PBS at 4 °C for 5 days and the dialysate was changed every 12 hours. The antigens were collected and centrifuged at 10,000× *g* for 5 min. The supernatant was obtained and stored in a −20 °C refrigerator for later use. Antigen synthesis was verified by 8453 UV–Visible spectrophotometer and the ratios of hapten/protein were estimated.

### 2.3. Immunization and Cell Fusion

All animal experiments were implemented in accordance with the animal ethics committee (HZAUMO-2021-0184) and following principles authorized by the Huazhong Agricultural University animal experiment center. Immunogens (A_1_, A_2_, A_3_) and two different antigen doses (50 µg and 100 µg) were inoculated into Female Balb/c mice, 6 weeks old. For the first immunization, immunogens were mixed with FCA and given subcutaneous multipoint injection in the back or neck of mice. Subsequently, the immunogens emulsified with FIA were used to enhance immunization every 14 days. The serum titer and sensitivity were measured by ic-ELISA. A mouse with high serum titer and sensitivity was selected and immunized with immunogens without Freund’s adjuvant for booster immunization. The mouse was euthanized and spleen cells were prepared for cell fusion. Activated SP2/0 myeloma cells and immunized spleen cells were fused at a ratio of 1:5~10 in 0.8 mL of 50% polyethylene glycol. The fusion cells were gently mixed in 1% HAT medium which contain feeder cells, then injected into 96-hole cell culture plates.

### 2.4. Generation of Hybridomas and Production of mAb

Hybridoma cells with highly sensitive ELISA positivity were cloned four times and expanded to screen the monoclonal cell line. Seven days later, the hybridoma cells (10^6^) stably secreting PYR and BaP antibodies were intraperitoneally injected into mice which had been cleared by 0.5 mL FIA. The mice were executed humanly 7–10 days later, and the ascites was gathered and stored at −80 °C.

### 2.5. Assessment of mAb

The subtype of mAb was determined by an antibody type identification kit. The performance of the mAb were assessed by measuring the IC_50_ values and cross-reactivity (CR). The CR for 16 PAHs was tested using PYR as the target analyte to estimate the specificity of the mAb. 

CR (%) = 100 × IC_50_ (PYR)/IC_50_ (other analogues). 

### 2.6. ELISA Procedure

The operation procedure of indirect ELISA (i-ELISA) described in Peng et al. was used with some modifications [26]. Microculture plates were added PBA-OVA coating antigen (100 µL/well), incubated at 4 °C overnight. After being three times washed by PBST (0.05% Tween 20 in PBS, pH 7.4), 200μL of OVA-PBS (1% OVA in PBS) was added into every well and blocked for 1 h at 37 °C. Washed again, 100 µL antibody was added into every well, and the plates kept for 40 min at 37 °C. Washed again, HRP-IgG (1:6000 dilution in 0.1M PBS, 100 µL/well) was added to all wells and the plates was kept at 37 °C for 40min. After being washed four times, TMB substrate solution (100 µL/well) was added. After incubation at 37 °C in the dark for 15 min, 50 µL of 1 M H_2_SO_4_ was added to finish the reaction. The absorbance at 450 nm was determined by an automatic ELISA reader. 

The operation procedure of indirect competitive ELISA (ic-ELISA) was in common with i-ELISA. The only difference was that the mAb and drug standard or sample (50 µL/well) were added to each well. The standard curve was established with logarithm of analyte concentration as the abscissa and the inhibition values (B/B_0_) corresponding to each concentration as the ordinate. 

### 2.7. Optimization of the ELISA Variables

#### 2.7.1. The Coating Antigen and mAb Titers

The optimum coating antigen concentration and mAb dilution were verified. A series of concentrations of PBA-OVA (1, 2, 4, 8 µg/L) were coated on the plates, and the mAb of multiple dilution was successively added. The standard curve was established by the ic-ELISA method. 

#### 2.7.2. The Working Concentration of HRP-IgG

The effect of the working concentration of HRP-IgG on the performance of ic-ELISA was evaluated. Ninety-six-well microculture plates were coated with the most optimal combination of coating antigen and mAb dilution. The sensitivity of ELISA to various working concentrations of HRP-IgG was tested for assay optimization, which were diluted to 1:4000, 1:5000, 1:6000 and 1:7000 with PBS. 

#### 2.7.3. Incubation Time

The sensitivity was also greatly affected by the incubation time of antigen and antibody. Under the above determined conditions, the competition time was set to 25, 35, 45 and 55 min; the HRP-IgG incubation time was set to 30, 40, 50 and 60 min to explore the optimal incubation time.

#### 2.7.4. Organic Solvents

To investigate the effect of different organic solvents on the binding ability of antigen and antibody, four solvents, DMF, acetone, acetonitrile and DMSO, were tested for assay optimization. Briefly, each solvent was diluted to 10%, 20%, 30% and 40% with PBS. Then, a series of PYR concentrations were prepared with the above dilution solvent, and standard curves were drawn according to the ic-ELISA steps.

### 2.8. Sample Preparation

Living aquatic products (fish, shrimp, crab) were obtained from local markets. The above samples were tested by HPLC-FLD and the negative samples were selected as blank samples. The pretreatment of the sample was as follows: the shell and viscera of fish, shrimp and crab samples were removed, three grams of homogenized samples were added into a 15 mL conical tube, 3 mL PBS buffer was added and agitated on a vortex mixer. Subsequently, 4 mL ethyl acetate and 2 mL acetonitrile were added, shaken thoroughly for 5 min and centrifuged at 5000× *g* for 5 min. The supernatant was blown to near dry with nitrogen. Next, 1 mL 20% DMF-PBS (1:4, v:v) was used to dissolved the residue, and then 1 mL n-hexane was added and shaken acutely, static for 5 min and remove supernatant. The subnatant was detected by ic-ELISA.

### 2.9. Validation Test of ic-ELISA

The ELISA validation was as follows: 20 aquatic products samples purchased from different areas were detected by ic-ELISA. The LOD and LOQ were calculated from the sum of the 20 blank samples. The precision and accuracy were evaluated by analyzing the recovery and intra- and inter-assay variations (CV) via the repeated analysis of the spiked samples. Briefly, three different concentrations of PYR (2, 4, and 8 µg/L) were spiked into the blank samples, and three intra-batch tests and five inter-batch tests were employed for ELISA. The recovery and CV were calculated as follows: recovery (%) = (conc. measured/conc. spiked) × 100; CV (%) = sample standard deviation/sample average × 100%.

### 2.10. Comparison of ELISA and HPLC-FLD Analysis

Compared with HPLC-FLD, the reliability of the ELISA method was verified by the same positive samples. The positive samples were homogenized and pretreated and detected by ELISA and HPLC-FLD. The HPLC-FLD method was performed according to Godinho et al. [25].

## 3. Results and Discussion

### 3.1. Antigen Verification and Serum Antibody Monitoring

The design and synthesis of antigens are very important for promoting the immune response of mice and preparing mAb with high-affinity and high-sensitivity. PYR is a small molecule compound that lacks functional groups conjugated with proteins and has no immunogenicity. In this study, pyrene butyric acid (PBA) was used as a hapten to synthesize artificial complete antigen. PAHs are composed of multiple benzene rings and have unique UV absorption spectra, so PBA hapten has multiple absorption peaks. The UV scanning spectra of immunogen PBA-BSA and coating antigen PBA-OVA are shown in Figure 1 and Figure 2.

As shown in Figure 1 and Figure 2, the UV absorption spectra of antigens changed significantly compared with those of carrier proteins (BSA, OVA) and PBA haptens and showed the characteristics of the absorption peaks of carrier proteins and haptens. The results showed that the PBA-BSA/OVA antigens were successfully synthesized. The estimated hapten/protein ratios of antigens were 8.1 (A_1_), 12.8 (A_2_), 18.3 (A_3_), 9.4 (B_1_), 11.7 (B_2_), and 19.2 (B_3_).

It has been reported that the immune effect of immunogens would be affected by the coupling rate of hapten and carrier proteins [27]. Therefore, we prepared antigens with different coupling ratios to explore their effects on the affinity and sensitivity of immunized mice. Three immunogens (A_1_, A_2_ and A_3_) were used to immunize mice. The absorbance values and inhibition rates of antiserum under different coupling ratio antigens are shown in Appendix A. It can be seen that in a certain range, the higher the hapten/protein ratio was, the better the serum titre and affinity of the antiserum were. This may be due to more haptens in the conjugate being exposed, which stimulated the mouse to produce more antibodies against PYR. Therefore, immunogen A_3_ and coating antigen B_3_ were the best combination.

### 3.2. Characterization of the mAbs

According to the antiserum results, the mouse with high sensitivity and inhibitory rate was selected for the cell fusion experiment. At present, PEG 4000 and PEG 1450 are commonly used for cell fusion. Due to the large molecular weight of PEG 4000, the cells lost water faster and fused more violently, causing cell injury. In contrast, the PEG 1450 fusion process was mild and achieved perfect results. After four subclones, PBA-BSA-A_3_ was used as the immunogen, and four specific mAbs against PYR and BaP were obtained. They were named 4A3, 4D6, 5D8 and 6H4, and the results are provided in Appendix A. The 4D6 mAb, which was selected for subsequent experiments because it exhibited the highest sensitivity, had an optimum dilution of 20,000 and an IC_50_ of 5.3 ng/mL. The results of mouse monoclonal antibody type identification indicated that the antibody subtypes were all IgG_1_ (Appendix A).

### 3.3. Optimization of the ELISA Variables

The concentrations of coating antigen and mAb, the incubation time of mAb and HRP-IgG, the working concentration of HRP-IgG and organic solvents were optimized, respectively. As shown in Appendix A, the optimal concentrations of coating antigen and mAb were 2 µg/mL and 1:20,000, respectively. The optimal incubation times of the mAb and HRP-IgG were 35 min and 40 min, respectively. The optimal working concentration of HRP-IgG was 1:6000.

PAHs are hydrophobic lipophilic substances that are usually dissolved in organic solvents. It has been reported that the parameters of ELISA would be influenced by the organic solvent [28]. Therefore, acetone, DMF, acetonitrile and DMSO were selected for study in this experiment. The effect of organic solvents added to the ELISA reaction system on the antigen-antibody binding reaction should be as small as possible. Therefore, it was necessary to explore the binding ability of different concentrations of organic solvents. Organic solvents with different volume fractions (10%, 20%, 30% and 40%) in PBS buffer were used as drug standard diluents. During the experiment, we found that acetone was highly volatile. When incubated at 37 °C, the total amount of liquid in the wells decreased significantly, which affected the accuracy of the results, so it was excluded.

As shown in Figure 3, compared with acetonitrile, the OD_450_ values of DMF and DMSO were relatively less affected. The use of organic cosolvents to improve the affinity and specificity of antibodies has also been supposed [29]. Then, the influence of concentrations of DMF and DMSO on the sensitivity of ELISA were investigated. When the volume fractions of DMF and DMSO were in the range of 10–40%, there was a large difference in sensitivity. In contrast, the IC_50_ value of 20% DMF-PBS (1:4, v:v) was the lowest, which significantly improved the sensitivity of ic-ELISA (Table 1). Therefore, 20% DMF-PBS (1:4, v:v) was selected as a drug standard diluent.

### 3.4. The Standard Curve and Cross-Reactivity

The PYR standard was diluted to 16.0, 8.0, 4.0, 2.0 and 1.0 µg/L. Under the above optimized conditions, the standard curve for PYR is shown in Figure 4. The equation was y = −44.558x + 76.59, ranging from 1 to 16 µg/L, R^2^ = 0.9929, and the IC_50_ was 3.73 ± 0.43 µg/L. 

The specificity of developed ic-ELISA for PYR mAb was assessed by CR. The standards containing 16 PAHs were tested for CR using 4D6 mAb. The mAb exhibited high binding affinity with PYR (100%) and BaP (38%), low affinity with fluoranthene (8%) and negligible cross-reactivity (< 1%) with other PAHs (Appendix A). This may be related to the structure of the hapten. The structures of PYR and BaP have the same four ring structures. Therefore, it is inevitable that antibodies prepared from PBA haptens cross with PYR and BaP. The CR results also confirmed our hypothesis. Previous studies on CR mainly focused on 2- to 5-ring PAHs [15,30,31]. However, this study mainly focused on 4- to 5-ring PAHs. Considering the CR for 16 PAHs, this mAb could specifically recognize PYR and BaP. Therefore, it can be used to set up an ic-ELISA kit to detect PYR and BaP in aquatic products.

### 3.5. Sample Preparation

In the establishment of the ELISA method, sample pretreatment is very important and is directly determined the accuracy of detection results. In recent years, the existing ELISA methods for PAHs detection have mainly concentrated on environmental samples, such as water (lake water, tap water, drinking water) and air [15,32,33]. However, there is no report on the detection of PAHs in aquatic products by ELISA. As a large country of aquatic products, it is imperative for China to detect PAHs in aquatic products. The ELISA method sensitivity is according to the specific binding ability of antigen and mAb, however, the sensitivity is susceptible to interference by fat and protein in the sample. Therefore, the samples usually need to be pretreated (extracted, purified, concentrated or diluted) before analysis to reduce matrix interference. To improve the precision of detection, according to the chemical properties, the extraction method of PYR and BaP in aquatic products was explored.

First, the matrix effect was evaluated, and blank samples were directly pretreated with PBS as previously reported [33]. The results showed that the matrix effect was very severe, even false-positive. When organic solvent extraction was performed during sample pretreatment, the matrix effect problem was solved. In this study, several extraction buffers, including acetonitrile, ethyl acetate and ethyl acetate-acetonitrile (4:1; v:v), were used to investigate the extraction effect. It can be seen that a higher recovery rate was obtained by ethyl acetate-acetonitrile (4:1; v:v). Acetonitrile could not extract the drug, and the recovery of ethyl acetate was less than 70%. Therefore, ethyl acetate-acetonitrile (4:1; v:v) was selected as the extraction solvent for sample pretreatment.

### 3.6. IC-ELISA Performance for Aquatic Products 

Based on the optimization of ic-ELISA conditions and sample pretreatment described above, 20 negative samples were collected for ELISA performance measurement. The LODs of the method for PYR and BaP in fish, shrimp and crab ranged from 0.43 to 0.54 µg/kg and 0.92 to 0.98 µg/kg, respectively. Meanwhile, the recoveries of PYR and BaP in fish, shrimp and crab were 81.5–101.9% and 84.9–94.0%, respectively, and the CV was less than 11.7% (Table 2). The minimum detection levels of PYR and BaP in aquatic products were far below the MRLs.

To validate the reliability of this experimental method, the ic-ELISA and HPLC-FLD methods were used to detect positive fish samples contaminated with PYR. The liquid chromatogram of PYR in the fish sample is shown in Appendix A. The results from the ic-ELISA were consistent with the results of HPLC-FLD, and the correlation coefficient was R^2^ = 0.9961 (Figure 5). The results indicated that ic-ELISA can be used as an available tool for the detection of PYR and BaP in aquatic products samples.

## 4. Comparison with Other Immunoassay Methods

Compared with other studies on immunoassay of PAHs, this study has the following advantages: firstly, this is the first study to establish an ELISA method for the detection of PYR and BaP in aquatic products, because most of the PAHS ELISA methods developed in recent years focus on environmental samples, such as water (lake water, tap water, drinking water) and air [15,32,33]. Secondly, although no new hapten was designed in this study, a monoclonal antibody with high sensitivity that can recognize PYR and BaP simultaneously was obtained by synthesizing complete antigens with different coupling ratios. Scharnweber et al. took BaP as the object to synthesize hapten. The procedure of this synthesis method is complicated, and hazardous chemical reagents such as toluene and pyridine are used in the synthesis of hapten, so it is difficult to control its harm to the environment and human beings in the synthesis process [29]. Finally, compared with the commercially available BaP kit of American REAGEN Company, the detection limit of this study is lower, and the pretreatment method is simpler.

## 5. Conclusions

In this study, antigens with different coupling ratios were synthesized and used to prepare a sensitive mAb, 4D6. The mAb showed high affinity for PYR (100%) and BaP (38%). Based on the mAb, a sensitive, efficient and reproducible ic-ELISA method was found and used for the determination of PYR and BaP residues in fish, shrimp and crab for the first time. The LODs of PYR and BaP in fish, shrimp and crab range between 0.43 and 0.98 µg/kg. Confirmation and relevance with HPLC-FLD showed the ic-ELISA method is a feasible tool. Therefore, the ic-ELISA method established in this experiment provided a practical tool for usual screening of PYR and BaP in aquatic products and supplemented the lack of an ELISA method in aquatic products. It lays a foundation for the development and application of PAHs residue detection kit in aquatic products. It has important academic value and practical significance to guarantee the quality of aquatic products.

## Figures and Tables

**Figure 1 foods-11-03220-f001:**
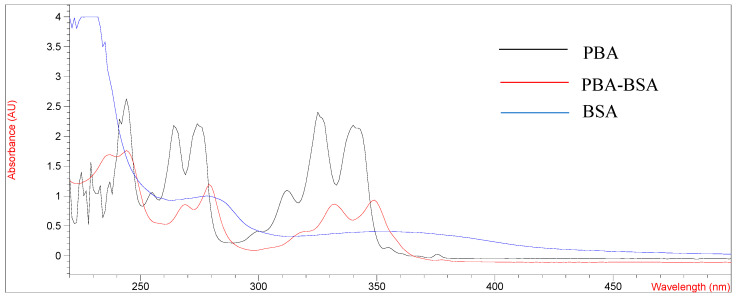
UV spectrum of BSA, PBA and PBA-BSA.

**Figure 2 foods-11-03220-f002:**
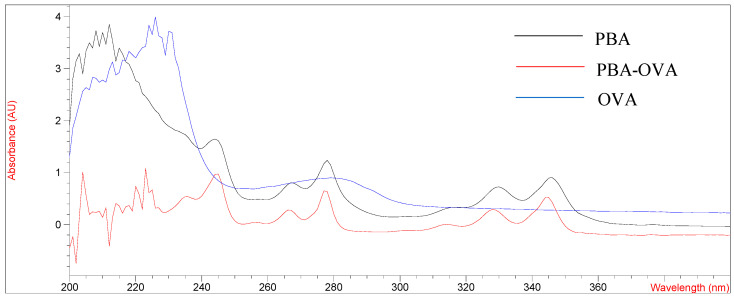
UV spectrum of OVA, PBA and PBA-OVA.

**Figure 3 foods-11-03220-f003:**
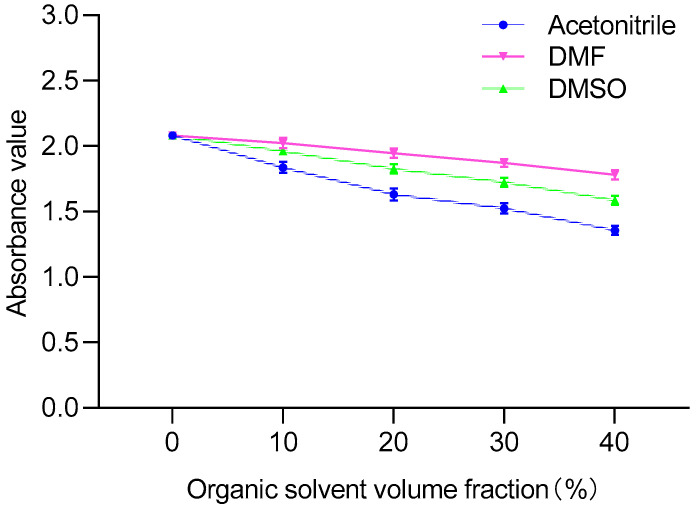
Comparison of different organic solvents.

**Figure 4 foods-11-03220-f004:**
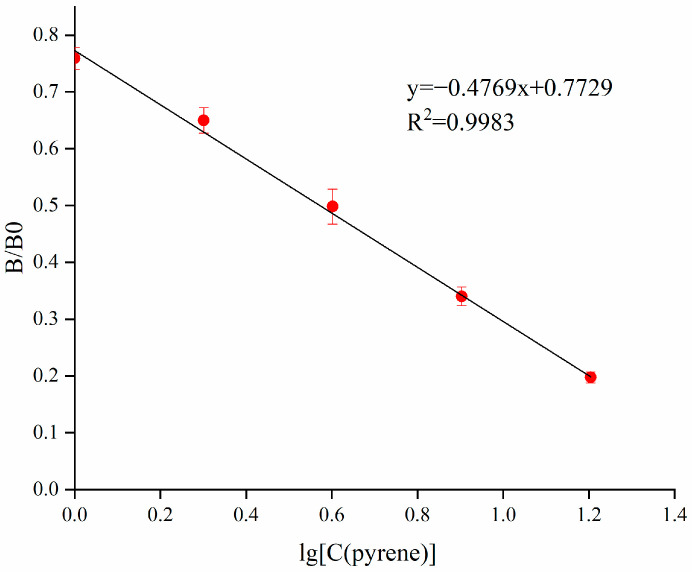
Standard curve for PYR in the ic-ELISA.

**Figure 5 foods-11-03220-f005:**
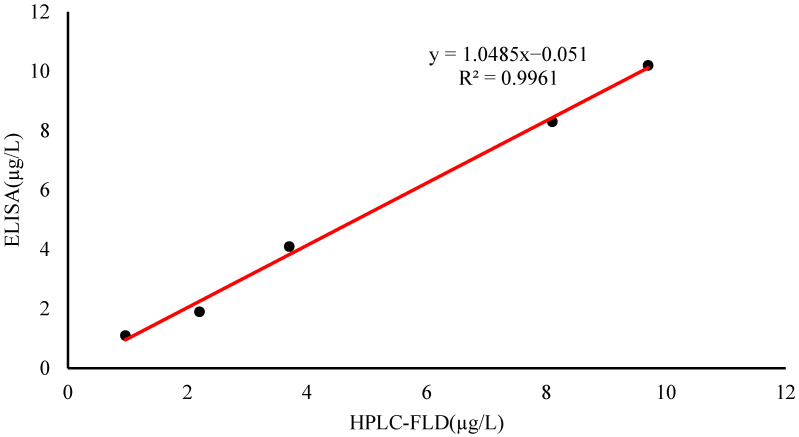
Correlation between ic-ELISA and HPLC-FLD for PYR determination of positive fish sample.

**Table 1 foods-11-03220-t001:** Comparison of different DMF and DMSO contents.

Organic Solvent (%)		Blank Hole OD_450_ Value	IC_50_ (µg/L)
DMF	10	1.872 ± 0.032	4.71 ± 0.31
	20	2.053 ± 0.031	3.37 ± 0.36
	30	2.113 ± 0.027	4.32 ± 0.28
	40	1.761 ± 0.041	3.91 ± 0.43
DMSO	10	1.972 ± 0.026	5.03 ± 0.37
	20	1.751 ± 0.043	5.74 ± 0.23
	30	1.721 ± 0.022	5.37 ± 0.28
	40	1.577 ± 0.034	6.62 ± 0.51

**Table 2 foods-11-03220-t002:** The LODs, LOQs, CV and recoveries of PYR and BaP in aquatic products.

Analytes	Samples	LOD (µg/L)	LOQ (µg/L)	Spiked Drug (ng/L)	Recovery (%)	CV (%) (n = 25)
PYR	Fish	0.54	0.69	2	81.5 ± 5.3	6.5
				4	93.7 ± 5.6	6.0
				8	86.3 ± 5.5	6.4
	Shrimp	0.43	0.56	2	91.1 ± 4.9	5.4
				4	84.3 ± 3.6	4.3
				8	101.9 ± 5.8	5.7
	Crab	0.53	0.71	2	99.4 ± 9.2	9.3
				4	86.3 ± 4.4	5.1
				8	82.3 ± 5.7	6.9
BaP	Fish	0.98	1.15	2	89.2 ± 5.5	6.2
				4	84.9 ± 6.1	7.2
				8	86.2 ± 4.9	5.7
	Shrimp	0.92	1.12	2	86.3 ± 6.1	7.1
				4	94.0 ± 6.5	6.9
				8	91.6 ± 8.6	9.4
	Crab	0.96	1.22	2	85.5 ± 4.3	5.0
				4	87.1 ± 5.7	6.5
				8	85.9 ± 7.5	8.7

## Data Availability

The data presented in this study are available on request from the corresponding author.

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
