# Peer review of "Preparation of Monoclonal Antibody against Pyrene and Benzo [a]pyrene and Development of Enzyme-Linked Immunosorbent Assay for Fish, Shrimp and Crab Samples"

_foods, 2022, doi:10.3390/foods11203220_

Round 1

Reviewer 1 Report

Title: Preparation of monoclonal antibody against pyrene and benzo[a]pyrene and development of enzyme-linked immuno-sorbent assay for aquatic products

Manuscript Number: foods-1914515

The manuscript entitled “Preparation of monoclonal antibody against pyrene and benzo[a]pyrene and development of enzyme-linked immuno-sorbent assay for aquatic products” is a more important study in the aquaculture industry. The authors have developed enzyme-linked immune sorbent assay with the help of monoclonal antibodies. This is an interesting work done by the authors. The manuscript needs to improve in English, and the information presented in this paper is detailed and precise. I recommend the authors undergo a thorough minor review of the manuscript for alignment corrections.

Abstract:

Author has to give types of aquaculture products (frozen or live animals).

Why does the industry need to check this pyrene and benzo[a]pyrene in aquaculture products? Is it just for export or import?

Provide more points about the outcome of this assay

2.8: What are all the organ samples taken from aquatic animals for this ELISA test?

What base author choused negative control for this test? Whether author confirmed with some other test?

The author needs to provide more details on the outcome and benefits of this study 

Author Response

Point 1: Author has to give types of aquaculture products (frozen or live animals).

Response 1: Thank you very much for your encouraging and helpful comments regarding our manuscript. We apologize for not describing clearly. We have further improved the part of abstract and sample preparation.

Point 2:Why does the industry need to check this pyrene and benzo[a]pyrene in aquaculture products? Is it just for export or import?

Response 2: Thank you very much for your encouraging and helpful comments regarding our manuscript. Firstly, the background part has introduced the misery and harm of pyrene and benzo pyrene in aquatic products. Moreover, the government has also issued corresponding documents, according to the national food safety standard GB 2762-2017, the limit of pollutants in food, benzo [a] pyrene in aquatic products, meat, cereals and oil products must be lower than the residue limit.

Point 3: 2.8: What are all the organ samples taken from aquatic animals for this ELISA test?

Response 3: Thank you very much for your encouraging and helpful comments regarding our manuscript. We have pointed out that the organ samples refer to residual tissue samples of fish, shrimp and crab after shelling and viscera removal.

Point 4:What base author choused negative control for this test? Whether author confirmed with some other test?

Response 4: Thank you very much for your encouraging and helpful comments regarding our manuscript. Firstly, HPLC-FLD method in GB 5009.265-2016 was used to verify aquatic product samples, and samples without pyrene and benzo [a] pyrene compounds were selected as negative samples. Then, we obtain positive control samples by adding quantitative standard compound to negative samples to establish our ELISA method.

Point 5:The author needs to provide more details on the outcome and benefits of this study 

Response 5: Thank you very much for your encouraging and helpful comments regarding our manuscript. We have supplemented the innovation and significance of this study in the "Comparison with other immutable methods" and “conclusion”.

Point 6:Provide more points about the outcome of this assay

Response 6: Thank you very much for your encouraging and helpful comments regarding our manuscript. Due to the space limitation, we put many results in the supplementary materials.

Reviewer 2 Report

The article titiled  Preparation of monoclonal antibody against pyrene and benzo[a]pyrene and development of enzyme-linked immuno-sorbent assay for aquatic products” presents the procedure of anti pyrene and benzopyrene monoclonal antibody preperation and the use of these antibodies  in an indirect competitive enzyme-linked immu-noassay (ic-ELISA) for  detection of PYR and BaP residues in aquatic products.

Below there are my comments:

1/ what is the reason that authors have prepared monoclonal antibodies against pyrene and benzo-pyrene if there are already published the procedures of such antibodies preparation and there are also possibility to buy commercial antibodies? What is new in the presented procedure of preparation in comparison with to the already published papers?

2/  authors have used the monoclonal antibodies prepared by them in ic-ELISA and detect PYR and BaP in fish, shrimp and crab. In my opinion the title should be changed and the names of specific aquatic products should be added. Please express in the titile what is new in this work,

3/ please correct figure 3, add SDs and add description of green curve; the same with figure 4 – please add SDs

Author Response

Point 1: What is the reason that authors have prepared monoclonal antibodies against pyrene and benzo-pyrene if there are already published the procedures of such antibodies preparation and there are also possibility to buy commercial antibodies? What is new in the presented procedure of preparation in comparison with to the already published papers?

Response 1: Thank you very much for your meaningful questions about our manuscript. As described in introduction, the residue of polycyclic aromatic hydrocarbons in aquatic products has been detected. Based on its harmfulness, we prepared a monoclonal antibody and an ELISA method was established for its residue detection to ensure food safety. Compared with other studies on immunoassay of PAHs, this study has the following advantages: firstly, this study is the first time to establish an ELISA method for the detection of PYR and BaP in aquatic products. Because most of the PAHS ELISA methods developed in recent years focus on environmental samples, such as water (lake water, tap water, drinking water) and air[15,32,33]. Secondly, although no new hapten was designed in this study, a monoclonal antibody with high sensitivity that can recognize PYR and BaP simultaneously was obtained by synthesizing complete antigens with different coupling ratios. Scharnweber et al. took BaP as the object to synthesize hapten. The procedure of this synthesis method is complicated, and hazardous chemical reagents such as toluene and pyridine are used in the synthesis of hapten, so it is difficult to control its harm to the environment and human beings in the synthesis process[29]. Finally, compared with the commercially available BaP kit of American REAGEN Company, the detection limit of this study is lower and the pretreatment method is simpler.

Point 2:Authors have used the monoclonal antibodies prepared by them in ic-ELISA and detect PYR and BaP in fish, shrimp and crab. In my opinion the title should be changed and the names of specific aquatic products should be added. Please express in the titile what is new in this work.

Response 2: Thank you very much for pointing out the shortcomings of our manuscript. We have changed the title as follows: Preparation of monoclonal antibody against pyrene and ben-zo[a]pyrene and development of enzyme-linked immuno-sorbent assay for fish, shrimp and crab samples.

Point 3:Please correct figure 3, add SDs and add description of green curve; the same with figure 4 – please add SDs.

Response 3: Thank you very much for pointing out the shortcomings of our manuscript. We have added SDs in figure 3 and figure 4, and described the curves with different colors.

Round 2

Reviewer 2 Report

Authors respond to all coments and manuscript is ready for publication in Foods.